# Performance of a new symptom checker in patient triage: Canadian cohort study

**Forson Chan**[1]\*, **Simon Lai**[2], **Marcus Pieterman**[1], **Lisa Richardson**[1], **Amanda Singh**[1], **Jocelynn Peters**[1], **Alex Toy**[1], **Caroline Piccininni**[1], **Taiysa Rouault**[2], **Kristie Wong**[1], **James K. Quong**[3], **Adrienne T. Wakabayashi**[1], **Anna Pawelec-Brzychczy**[1]

**1** Department of Family Medicine, Schulich School of Medicine and Dentistry, Western University, WCPHFM, London, ON, Canada, **2** University of British Columbia, Faculty of Medicine, Health Sciences Mall, Vancouver, Canada, **3** Whitehorse General Hospital, Whitehorse, YT, Canada

\* forsonc@alumni.ubc.ca

## Abstract

### Background

Computerized algorithms known as symptom checkers aim to help patients decide what to do should they have a new medical concern. However, despite widespread implementation, most studies on symptom checkers have involved simulated patients. Only limited evidence currently exists about symptom checker safety or accuracy when used by real patients. We developed a new prototype symptom checker and assessed its safety and accuracy in a prospective cohort of patients presenting to primary care and emergency departments with new medical concerns.

### Method

A prospective cohort study was done to assess the prototype's performance. The cohort consisted of adult patients ($\geq$16 years old) who presented to hospital emergency departments and family physician clinics. Primary outcomes were safety and accuracy of triage recommendations to seek hospital care, seek primary care, or manage symptoms at home.

### Results

Data from 281 hospital patients and 300 clinic patients were collected and analyzed. Sensitivity to emergencies was 100% (10/10 encounters). Sensitivity to urgencies was 90% (73/81) and 97% (34/35) for hospital and primary care patients, respectively. The prototype was significantly more accurate than patients at triage (73% versus 58%, $p$<0.01). Compliance with triage recommendations in this cohort using this iteration of the symptom checker would have reduced hospital visits by 55% but cause potential harm in 2–3% from delay in care.

### Interpretation

The prototype symptom checker was superior to patients in deciding the most appropriate treatment setting for medical issues. This symptom checker could reduce a significant

**Data Availability Statement:** The majority of the relevant data is within the paper and its Supporting Information files. The current information provided in the supplemental should be enough for any interested reviewer to decide if the data was

interpreted appropriately. Sensitive patient information (i.e. written by treating physicians) will be available with approval from the Western University Research Ethics Board or Lawson Health Research for researchers who meet the criteria for access to confidential data. The contact for Western Research and Lawson is provided below, as they are the most responsible for safeguarding patient information in this study. Western Research Room 5150 Support Services Building, 1393 Western Road London, Ontario, Canada, N6G 1G9 Tel: 519-661-2161 | Research Ethics: 519-661-3036 res-serv@uwo.ca Lawson Health Research 750 Base Line Road East, Suite 300 London, Ontario, Canada N6C2R5 Tel: 519-667-6649 Email: info@lawsonresearch.com.

**Funding:** FC received the Unnur Brown Leadership Award in Health Policy, which was granted by the Dr. Adalsteinn Brown and the Larry and Cookie Rossy Family Foundation and the Schulich School of Medicine and Dentistry (schulich.uwo.ca). FC also received a Resident Research Grant from the PSI Foundation (www.psifoundation.org). The funders had no role in study design, data collection and analysis, decision to publish, or preparation of the manuscript.

**Competing interests:** The authors have declared that no competing interests exist.

number of unnecessary hospital visits, with accuracy and safety outcomes comparable to existing data on telephone triage.

## Background

Worldwide, 70% of internet users go online to access health information [1–3]. Most inquiries begin with search engines such as Google, but results are often incomplete, inaccurate, or inaccessible to lay persons [3–5]. Potential consequences of unreliable or inappropriate health information include delay of care, inappropriate hospital visits, and cyberchondria [6, 7]. These online health-seeking behaviors may be driven in part by a knowledge gap with respect to what people should do if they become ill, given that public health education focuses primarily on preventive health [8, 9]. This knowledge is also difficult to teach, given that even trained health professionals may struggle to identify patients who would most benefit from emergency department care [9–13].

During the COVID-19 pandemic, computerized algorithms known as "symptom checkers" were widely implemented to limit unnecessary contact between patients and healthcare providers, alleviate pressures on telehealth systems, and empower patients to decide where best to access care for their symptoms [14–18]. Symptom checkers are not new, and many already existed prior to the pandemic [19]. Unfortunately, few studies have been published about the safety or accuracy of symptom checkers and, of existing studies, all suffer from at least one of the following limitations: the literature was not peer-reviewed; the results were based solely on simulated data, the selection of patients covered a limited range of conditions; the data was not generated by patients entering in their own symptoms; or the studies did not report safety outcomes [20]. Development of a safe and effective symptom checker could lower barriers to accessing care, encourage patients to go to the hospital for potentially life-threatening problems, discourage unnecessary healthcare visits, and reduce wait times by encouraging appropriate healthcare utilization [19, 21–25].

We developed our own prototype symptom checker [26] and conducted this study to address the methodological limitations found in existing studies [27, 28]. In particular, we detail the performance of the symptom checker when used by adults seeking care from family physicians and emergency departments.

## Methods

### Development of the symptom checker

The prototype symptom checker was designed by author FC and coded by computer science students at Western University [26]. The version of the algorithm used for this study was comprised of a total of 247 questions or computational steps manually compiled into a decision tree with recursive elements. Relevant questions are presented sequentially to the user based upon previous responses.

To create the questions, common medical diagnoses and concerns were first collected through medical textbooks (e.g. Toronto Notes, 2018; Edmonton Manual, 2018; Harrison's Internal Medicine 20e) and online physician resources (e.g. Uptodate, Dynamed, College of Family Physicians of Canada). Literature review was then performed for each diagnosis to identify questions and factors that would impact escalation of treatment. Diagnostic questions were specifically gathered for conditions considered emergency and conditions that can be managed by patients at home. Diagnostic questions that did not change triage

recommendations were discarded. No patient vignettes were used for the development of this symptom checker.

## Study design

A prospective cohort study was done with patients who self-presented to emergency departments and primary care clinics. Ethics approvals were obtained from the research ethics boards of Western University, the University of British Columbia, and Whitehorse General Hospital, respectively. The prospective sample included Canadian patients presenting to 2 emergency departments (Whitehorse General Hospital in Whitehorse, YT; Royal Inland Hospital in Kamloops, BC) and 13 full-time family physician practices based out of 3 centres in London, Ontario (Victoria Family Medical Centre, St. Joseph's Family Medical Centre, and Byron Family Medical Centre). The research ethics committees approved the lack of parent or guardian consent for minors because all intended participants are 16 years of age or older. The sample was representative of adult Canadian patients who seek professional medical attention.

In the waiting rooms of each location, an unattended kiosk and information poster invited adult patients to use a tablet computer programmed with the prototype symptom checker. Inclusion criteria were adults ≥16 years reporting that they were seeking care for a new medical issue. The information poster contained details typically found in a patient letter of information. Patients were incentivized to participate with opportunities to win $50 gift cards. On the tablets, patients signed consent and entered their name, age, gender, and symptom checker responses. Patient recruitment and data collection occurred from June 2019 to February 2020. Patients were blinded to the symptom checker's triage recommendation after completion and treating physicians were blinded to patient participation.

## Data management

Identifying information and responses to the prototype symptom checker were stored in separate files on the tablets to ensure blinding during analysis. Identifying information was used to link medical records. If the corresponding patient records could not be found, the symptom checker responses were excluded from analysis. A list of valid medical records was collected and authors (MP, LR, AS, JP, AT, CP, TR) extracted the impression and plan written by the treating physician.

The diagnoses and treatments provided by the treating physician were reviewed by physician authors FC and SL and, by consensus, assigned an appropriate categorization for comparison with triage recommendations provided by the symptom checker (Table 1). The process of dividing patient encounters into broad categories of immediate/emergent, urgent, primary/routine care, and home/self-care is a standard process in similar studies that examine the

Table 1. Criteria used for categorizing the acuity of patient medical encounters with a healthcare professional.

| Information from patient medical records | Subsequent categorization of the patient | Most appropriate triage location for this categorization |
|---|---|---|
| Patient was admitted to hospital, sent to hospital, or required immediate treatment usually only available in hospital. | Emergency | Hospital |
| Patient's diagnosis was not life or limb threatening, but required timely assessment by an emergency physician or primary care provider for treatment or referral for treatment | Urgency | Hospital or primary care |
| Initial treatment provided was wholly within scope of an outpatient family physician practice. Minimal risk for significant harm would result from delay in providing the treatment administered by the physician. | Routine | Primary care |
| Management advice was given to the patient for an issue that did not require a prescription, referral, or further diagnostic testing. | Home- Appropriate | Home |

performance of telemedicine triage and emergency triage systems [29, 30]. Categorization was done with blinding to triage recommendations made by the symptom checker. The treating physicians' diagnoses and management plans were used the gold standard for comparison with recommendations provided by the symptom checker; patients' presenting symptoms were not considered when categorizing the true severity of the patient's medical issue.

Some physicians documented multiple diagnoses or issues for the healthcare visit. In these cases, each diagnosis was separately categorized as an emergency, urgency, routine, or home appropriate. For example, a patient encounter with dual diagnoses of "1) vaginal discharge not yet diagnosed and 2) lower back pain" were categorized as routine and home appropriate, respectively, because investigations were ordered by the physician for the vaginal discharge and home-treatment solutions were recommended for the lower back pain. Symptom checker responses were then assessed to determine the most appropriate physician diagnosis. In this specific encounter, the patient selected on the symptom checker that the concern was not related to pain; thus the diagnosis "vaginal discharge not yet diagnosed" was used for the purposes of assessing symptom checker performance.

Patients were excluded from final analysis if the treating physician commented that the patient was being seen for follow-up, or if there was a very significant mismatch between patient and physician concerns for visit (e.g. the patient selected "I want to seek help related to violence against women" and the visit diagnosis was "otitis media").

## Statistical methods

Based on data gathered in the first month, study recruitment to allow for 80% power was guided by a predicted 20% prevalence and a conservative 75% sensitivity for urgencies. 234 participants were required from each of hospital and clinic settings. To adjust for patient records that could not be identified, minimum recruitment was set at 292 participants.

Outcomes measures were pre-specified. Triage was considered accurate if the triage categorization, as determined by the treating physician's management plan, matched the triage recommendation provided by the symptom checker. Data were calculated with 95% confidence intervals about the mean. Sensitivity and positive predictive value (PPV) were calculated with 95% confidence intervals (Wilson-score method). A two-tailed McNemur's Test was used to compare the accuracy of patient decision making with the prototype. Calculations were done with Microsoft Excel (2016) and SPSS (version 21).

## Results

Included for final analysis were 281 and 300 patients who presented to emergency departments and primary care clinics, respectively (Fig 1). Mean patient age was 38±16 years (range 16 to 91 years) for emergency department patients and 48±18 years (range 16 to 91 years) for patients seen by primary care family physician. 366 of 581 patients (63%) were female. There was a diverse collection of patient concerns captured including trauma, mental health, pregnancy, immunizations, infections, cardiovascular, respiratory, gastrointestinal, and dermatologic concerns, among others. Of patients with mismatches between patient and physician responses, mean age was 40±15 years (range 19 to 94 years) for emergency patients and 50±20 years (range 19 to 94 years) for patients seen by primary care. The algorithm's triage "accuracy" for these mismatched cases was 90%, but was not included for analysis because the patient used the symptom checker for an entirely different reason than what was discussed with the physician.

For emergency department patients (Table 2; n = 281), the accuracy of the prototype symptom checker (172/281 encounters or 61%; 95% CI of 55% to 67%) was significantly better than

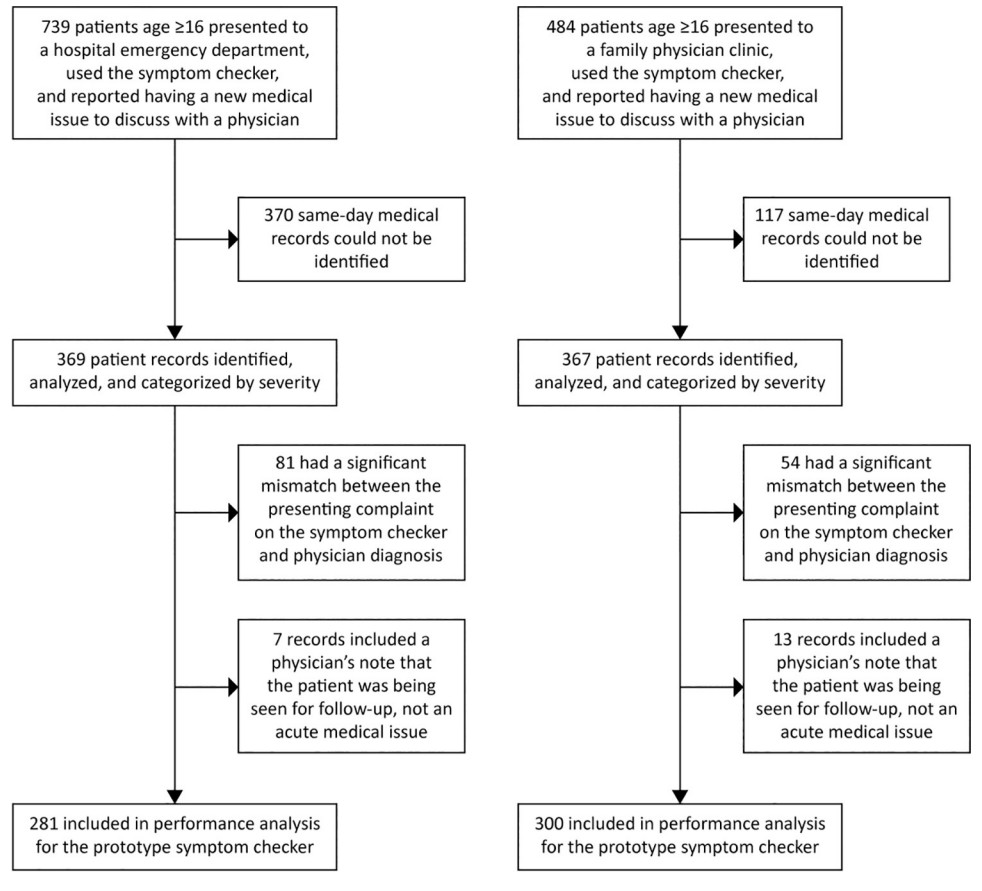

**Fig 1. Flow of study participants.**

that of patients (90/281 encounters, 32%; CI of 27% to 38%; $p < 0.01$). No emergencies were missed by the symptom checker (n = 8); the diagnoses of these emergencies were: ingestion of death camas; incarcerated inguinal hernia; biceps tendon tear or rupture; alcohol intoxication with coffee ground emesis; upper gastrointestinal bleed requiring transfusion; intoxication, chest pain, and possible Brugada; self-inflicted wrist laceration; hand laceration from a drill

**Table 2. Triage recommendations made by the prototype symptom checker for patients self-presenting to hospital.**

| | | Encounter categorization (n = 281) | | | | |
| --- | --- | --- | --- | --- | --- | --- |
| | | Emergency | Urgency | Routine | Home-Appropriate | PPV |
| Symptom checker's management recommendation | Hospital | 9 | 41 | 54 | 22 | 40% (32–48%) |
| | Primary Care | 0 | 32 | 78 | 8 | 93% (87–97%) |
| | Home | 0 | 8 | 17 | 12 | 32% (0–14%) |
| | Event rate | 3% | 29% | 53% | 15% | |
| | Sensitivity | 100% | 90% | 52% | 29% | |
| | | (70–100%) | (82–95%) | (44–60%) | (17–44%) | |

Green colored cells represent accurate triage recommendations made by the symptom checker. Red colored and uncolored cells represent under-triage and over-triage, respectively. Sensitivity and positive predictive value (PPV) are reported with 95% confidence intervals).

**Table 3. Triage recommendations made by the prototype symptom checker for patients self-presenting to family physician clinics.**

| | | Encounter categorization (n = 300) | | | | |
| --- | --- | --- | --- | --- | --- | --- |
| | | **Emergency** | **Urgency** | **Routine** | **Home-Appropriate** | **PPV** |
| Symptom checker's management recommendation | Hospital | 1 | 26 | 15 | 5 | 57% (43–70%) |
| | Primary Care | 0 | 8 | 193 | 18 | 88% (83–92%) |
| | Home | 0 | 1 | 6 | 27 | 79% (63–90%) |
| | Event rate | 0.3% | 12% | 71% | 17% | |
| | Sensitivity | 100% | 97% | 90% | 54% | |
| | | (21–100%) | (85–99%) | (85–93%) | (40–67%) | |

Green colored cells represent accurate triage recommendations made by the symptom checker. Red colored and uncolored cells represent under-triage and over-triage, respectively. Sensitivity and positive predictive value (PPV) are reported with 95% confidence intervals).

through the hand; and antepartum hemorrhage. Sensitivity to urgencies was 90% (73/81 encounters; CI of 82% to 95%). Under-triage for urgencies occurred in 3% (8 encounters, CI of 1% to 6%); diagnoses for these cases were: abscess of thigh, phalanx fracture, boxer's fracture, post lipoma excision hematoma, hand laceration, metacarpal fracture, toe fracture, and metallic foreign body in shin. Over-triage to hospital occurred in 76 encounters (27%, CI 22% to 33%). Patient compliance with the symptom checker's triage advice would have reduced total hospital visits by 55% (155/281 encounters, CI of 49% to 61%).

For patients who presented to primary care (Table 3; n = 300), the accuracy of the symptom checker (85% or 255/300 encounters; CI of 81%-89%) was similar to that of patients' (83% or 249/300; CI of 78% to 87%; $p$ = 0.11). One patient with an emergency diagnosis of "possible deep vein thrombosis" presented to primary care, but the symptom checker would have correctly advised the patient to go to the hospital. Sensitivity to urgencies was 97% (34/35, CI 85% to 99%). Under-triage occurred in one urgency encounter wherein the diagnosis was "possible cellulitis surrounding tracheostomy". Over-triage to hospital occurred in 20 encounters (7%, CI 4% to 10%) in which hospital triage was suggested for issues that could be managed either at home or by primary care.

The overall accuracy of the symptom checker was 73% (427/581, 95% CI 70% to 77%), which was significantly better than the 58% accuracy of patients (339/581, CI 54% to 62%; $p<0.01$). Under-triage causing potential harm from delay in care occurred in a total of 9 encounters (2%, CI 1% to 3%), in which the symptom checker recommended home management was for an urgency.

## Discussion

This is the first study to report a direct comparison between the triage accuracy of a symptom checker against decisions made by patients. The overall accuracy of our prototype symptom checker was 73% (95% CI 70% to 77%), with potential to harm in 2–3% of encounters. These results are promising given that, in context, telephone triage is reported to have a median accuracy 75% and a rate of harm from under-triage of 1.3–3.2% [29]. The results of this study may be most directly applied to a scenario wherein presenting patients at healthcare facilities have an opportunity to obtain a rapid computer generated opinion about their medical concern, with the possibility of redirecting care to a nearby hospital or clinic providing primary care for walk-in patients.

The symptom checker was less accurate among patients who presented to hospital compared to those who presented to primary care. Over-triage to hospitals occurred more often for patients who self-presented to the emergency department (76/281, 27%) compared to those

who sought primary care (20/300, 7%). This may have occurred because patients who go to the hospital reported having more severe symptoms when using the symptom checker, which resulted in the symptom checker recommending a higher acuity response. Under-triage for urgencies occurred in several instances specifically related to distal extremity injuries and improvements will need to be made for these types of presenting concerns. The physician visit diagnosis and patient's symptom checker responses were significantly mismatched for 135 patients and excluded from analysis (S1 Table); this may have been because patients wanted to utilize the symptom checker out of personal interest, but did not wish to reveal details about their personal health information.

This is the first study to fulfill three key criteria in the assessment of a symptom checker's triage performance which was found to be lacking in previous studies: triage recommendations are compared to care provided by a clinician as the reference standard; testing was done in a general population of patients who continue to receive standard care; and an unrestricted range of symptoms was assessed [27, 28]. All peer-reviewed studies involving patients published thus far do not meet one or more of the above criteria (Table 4) [25, 31–39]. Three non-peer reviewed reports were reviewed; two had significant risks for bias and the one government report did not provide sufficient information to interpret outcome measures [40–42].

## Limitations

Across emergency department and primary care settings, the prototype appropriately triaged 10 of 10 patients diagnosed with an emergency but the study was not powered for emergencies. The sample size of emergencies was small because of the relative infrequency of emergencies compared to other medical presentations. Emergencies were also more likely to arrive by ambulance which bypasses the waiting room. Patients who felt very unwell were also less likely to use the symptom checker, based on the study design. Further studies will be needed to ensure sensitivity to emergencies.

Reproducibility of the study's results in other countries may be challenging given that local guidelines, cultural factors, and access to resources may differ. The results of these studies cannot be generalized to other symptom checkers given the diversity of triage approaches [19].

**Table 4. Summary of peer-reviewed papers on symptom checker usage by patients.** All studies were limited by at least one of three factors (in grey).

| Paper | Assessment of Triage Accuracy | Population | Symptoms assessed |
|---|---|---|---|
| Meyer et al.[23] | Did not assess accuracy | General population of registered users of the symptom checker | Unrestricted |
| Berry et al.[31] | Assessed accuracy | Patients presenting to an outpatient internal medicine clinic with abdominal pain symptoms | Abdominal pain only |
| Nijland et al.[32] | Did not assess accuracy | General population of web users | Unrestricted |
| Poote et al.[33] | Risks of bias in assessing accuracy (Treating physicians were not blinded to triage outcomes, potentially subjective triage categorization schema) | University students only | Unrestricted |
| Verzantvoort et al.[34] | Accuracy was measured using nursing triage as the reference standard. Triage does not assess potential emergencies in determination of accuracy | General population of primary care patients | Unrestricted |
| Sole et al.[35] | Only 5% (n = 59) of symptom checker users were able to be assessed by a physician for accuracy | College students only | Unrestricted |
| Price et al.[36] | Assessed accuracy | Children only | Influenza-like illness only |
| Winn et al.[37] | Did not assess accuracy | Users of an online chat bot | Unrestricted |
| Cowie et al.[38] | Did not assess accuracy | General population in Scotland | Unrestricted |
| Miller et al.[39] | Did not assess accuracy | General population in England | Unrestricted |

Some questions remain unanswered by this study. It is unclear if patients will respond to recommendations by the symptom checker. It is also uncertain how the symptom checker would perform among a population of patients who have chosen to stay at home, instead of seeking care at a hospital or outpatient clinic. Implementation of a symptom checker on a community level would be necessary to demonstrate if implementation results in changes in healthcare utilization and changes in population morbidity.

## Conclusions

The prototype symptom checker was superior to patients in deciding the most appropriate treatment setting for medical issues. Use of the symptom checker by patients seeking medical care could reduce a significant number of unnecessary hospital visits, with accuracy and safety outcomes comparable to existing data on telephone triage.

## Supporting information

**S1 Table. Raw data: Comparison of symptom checker recommendations to physician diagnosis and treatment.** De-identified data containing information ported directly from patient charts may be available upon approval from the Western Research Ethics Board for authorized individuals.
(XLSM)

## Acknowledgments

This study was done with the cooperation of the staff at London Health Science Centre's family medical centres (Victoria Family Medical Centre and Byron Family Medical Centre), St. Joseph's Family Medical and Dental Centre, Whitehorse General Hospital, and Royal Inland Hospital. The prototype symptom checker app was programmed by Sama Rahimian, Brandon Kong, Zenen Treadwell, and Hussein Fahmy. We thank all the physicians, custodial staff, administrative staff, and study patients for help with planning, accommodating the needs of this study, and sampling. In particular, we thank Dr. Ian Mitchell, Dr. Sonny Cejic, Dr. Saadia Hameed, Dr. Evelyn Vingilis, Lindsey Page, and Mike Rickson for their support.

## Author Contributions

**Conceptualization:** Forson Chan, Simon Lai, Adrienne T. Wakabayashi, Anna Pawelec-Brzychczy.

**Data curation:** Forson Chan, Simon Lai, Marcus Pieterman, Lisa Richardson, Amanda Singh, Jocelynn Peters, Alex Toy, Caroline Piccininni, Taiysa Rouault.

**Formal analysis:** Forson Chan, Simon Lai, Marcus Pieterman, Lisa Richardson, Amanda Singh, Jocelynn Peters, Alex Toy, Caroline Piccininni, Taiysa Rouault.

**Funding acquisition:** Forson Chan, Anna Pawelec-Brzychczy.

**Investigation:** Forson Chan, Simon Lai, James K. Quong, Adrienne T. Wakabayashi, Anna Pawelec-Brzychczy.

**Methodology:** Forson Chan, Simon Lai, James K. Quong, Adrienne T. Wakabayashi, Anna Pawelec-Brzychczy.

**Project administration:** Forson Chan, Simon Lai, Marcus Pieterman, Lisa Richardson, Amanda Singh, Jocelynn Peters, Alex Toy, Caroline Piccininni, Taiysa Rouault, James K. Quong, Adrienne T. Wakabayashi, Anna Pawelec-Brzychczy.

**Resources:** Forson Chan, Adrienne T. Wakabayashi.

**Software:** Forson Chan.

**Supervision:** Forson Chan, Simon Lai, James K. Quong, Anna Pawelec-Brzychczy.

**Validation:** Forson Chan, Simon Lai, Marcus Pieterman, Lisa Richardson, Amanda Singh, Jocelynn Peters, Alex Toy, Caroline Piccininni, Taiysa Rouault.

**Writing – original draft:** Forson Chan, Marcus Pieterman, Lisa Richardson, Jocelynn Peters.

**Writing – review & editing:** Forson Chan, Simon Lai, Marcus Pieterman, Lisa Richardson, Amanda Singh, Jocelynn Peters, Alex Toy, Caroline Piccininni, Taiysa Rouault, Kristie Wong, James K. Quong, Adrienne T. Wakabayashi, Anna Pawelec-Brzychczy.

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
