## [Decision Letter · Decision Letter 0]

26 May 2021

PONE-D-20-39920

Performance of a New Symptom Checker in Patient Triage: Canadian Cohort Study

PLOS ONE

Dear Dr. Chan,

Thank you for submitting your manuscript to PLOS ONE. After careful consideration, we feel that it has merit but does not fully meet PLOS ONE’s publication criteria as it currently stands. Therefore, we invite you to submit a revised version of the manuscript that addresses the points raised during the review process.

The reviewers have identified several aspects of your study design that require further clarification. Please ensure that you address these points thoroughly in your revisions.

We look forward to receiving your revised manuscript.

Kind regards,

Jamie Males

Staff Editor

PLOS ONE

Journal Requirements:

2. Please state in your methods section whether you obtained consent from parents or guardians of the minors included in the study or whether the research ethics committee or IRB approved the lack of parent or guardian consent.

3. In your Methods section, please provide additional information about the participant recruitment method and the demographic details of your participants. Please ensure you have provided sufficient details to replicate the analyses such as: a) the recruitment date range (month and year), b) a description of any inclusion/exclusion criteria that were applied to participant recruitment, c) a table of relevant demographic details, d) a statement as to whether your sample can be considered representative of a larger population, e) a description of how participants were recruited, and f) descriptions of where participants were recruited and where the research took place.

4.We note that you have indicated that data from this study are available upon request. PLOS only allows data to be available upon request if there are legal or ethical restrictions on sharing data publicly. For information on unacceptable data access restrictions, please see http://journals.plos.org/plosone/s/data-availability#loc-unacceptable-data-access-restrictions.

Reviewers' comments:

Reviewer's Responses to Questions

**Comments to the Author**

1. Is the manuscript technically sound, and do the data support the conclusions?

Reviewer #1: Partly

Reviewer #2: Yes

2. Has the statistical analysis been performed appropriately and rigorously? 

Reviewer #1: Yes

Reviewer #2: Yes

3. Have the authors made all data underlying the findings in their manuscript fully available?

Reviewer #1: No

Reviewer #2: Yes

4. Is the manuscript presented in an intelligible fashion and written in standard English?

Reviewer #1: Yes

Reviewer #2: Yes

5. Review Comments to the Author

Reviewer #1: A good small proof of concept study to assess a computerized symptom checker with actual care provided in 2 emergency departments (Whitehorse and Kamloops, BC) and 13 full-time family physicians in London.

Nice concept but lack details and has the several weaknesses

1. Very small sample size evident from large confidence intervals of the reported results

2. Inter-rater reliability of physician reviewer for categorization

3. Very scant details regarding the algorithms created

4. While the checker performs in a reasonable way for identifying emergencies (with wide confidence intervals due to small sample size), its overall accuracy of around 60% is a concern.

Reviewer #2: This study addresses a major gap in the evaluation of symptom checkers (SC), the use by real patients at the time of presentation with a medical complaints in the ED or urgent primary care with results compared to actual clinical decisions. The authors developed their own symptom checker to advise on triage and appropriate level of care. This was tested in two hospital emergency departments and in 13 primary care sites in Canada. Patients and clinicians were blinded to the symptom checkers responses. Records were linked from patients name, age and gender. Urgency of triage was assessed with a 4 point scale (immediate/emergency, urgent, primary/routine, home care).

Results show that the new SC performed equivalently in sensitivity and PPV to telephone triage (based on a systematic review from 2012). No emergency cases were missed but 8 patients in hospital with the need for urgent care would have been recommended to stay home. One patient in primary care with an urgent problem would have similarly been told to stay home. In the hospital context the SC was significantly more accurate at triage than the patients themselves, but in primary care accuracy was very similar. The only significant weakness noted is a lack of information on the patient’s experience in using the SC.

Overall this is a very good study that is much larger and higher quality that nearly all existing studies. It has a low risk of bias, based on accepting all ED or primary care patients, having patients enter data on their own symptoms and comparing the SC performance to the actual physicians’ decisions. This study makes a major contribution to the field.

Minor revisions:

Clarification is required in the handling of cases where there was more than 1 diagnosis. It is stated that “Symptom checker responses were then assessed to determine the most appropriate physician diagnosis.” What were the criteria for determining the appropriate one – whether the data collected by the symptom checker was more supportive of one of the physician’s diagnoses? What if the data included questions relevant to both? What number/percentage of cases had more than one physician diagnosis?

Patient were excluded “if there was a very significant mismatch between patient and physician concerns for visit”. What number/percentage of cases were excluded by this criteria?

In listing the 3 key criteria for quality of studies the authors may wish to add a 4th, data entered by patients on their own symptoms, which is missing from most existing studies of SCs, that use existing or made up cases.

6. PLOS authors have the option to publish the peer review history of their article (what does this mean?). If published, this will include your full peer review and any attached files.

Reviewer #1: No

Reviewer #2: **Yes: **Dr Hamish Fraser

---

## [Author Response · Author response to Decision Letter 0]

13 Oct 2021

Reviewer #1: A good small proof of concept study to assess a computerized symptom checker with actual care provided in 2 emergency departments (Whitehorse and Kamloops, BC) and 13 full-time family physicians in London.

Nice concept but lack details and has the several weaknesses

1. Very small sample size evident from large confidence intervals of the reported results

2. Inter-rater reliability of physician reviewer for categorization

3. Very scant details regarding the algorithms created

4. While the checker performs in a reasonable way for identifying emergencies (with wide confidence intervals due to small sample size), its overall accuracy of around 60% is a concern.

Response to Reviewer #1:

1) Contextually in this field of study, the sample size in our study is actually quite large (n=581). Of existing studies on symptom checkers that assessed for triage accuracy when used patients (Table 4), the sample sizes were 49 (Berry et al.) and 294 (Price et al.) individuals. As a result of the large sample size, the confidence intervals were quite narrow except for encounters for medical emergencies, which is likely what the reviewer has issue with. The sample size for emergencies was small, as expected (n=10). This is also addressed under the limitations section of the manuscript, which is quoted here for convenience: “Across emergency department and primary care settings, the prototype appropriately triaged 10 of 10 patients diagnosed with an emergency but the study was not powered for emergencies. The sample size of emergencies was small because of the relative infrequency of emergencies compared to other medical presentations. Emergencies were also more likely to arrive by ambulance which bypasses the waiting room. Patients who felt very unwell were also less likely to use the symptom checker, based on the study design. Further studies will be needed to ensure sensitivity to emergencies.”

2) Inter-rater reliability data is not available because it was done by consensus between FC and SL. This detail is now make more clear in the Methods section.

3) The purpose of this paper is not to detail how the algorithm functions, but how well it functions. Contextually, in most studies assessing the accuracy of patient telephone triage, the information provided is even more limited regarding the protocols in place to triage patients and compliance with operational standards. In addition, the algorithm has been made available to be used for free online at symptomcheck.ca, as cited in the manuscript.

4) The overall accuracy of the symptom checker was 73%. Contextually, telephone triage has a similar level of accuracy, as stated in the Discussion section. In the sub-population of patients from emergency departments, the accuracy was lower (61%) because it was more likely to recommend patients to go to the hospital for routine issues, likely because they reported more severe symptoms (see 2nd paragraph in the discussion); this was still much more accurate than patient decision making. The algorithm correctly identified 10 of 10 medical emergencies in the sample size.

Reviewer #2:

Minor revisions:

1) Clarification is required in the handling of cases where there was more than 1 diagnosis. It is stated that “Symptom checker responses were then assessed to determine the most appropriate physician diagnosis.” What were the criteria for determining the appropriate one – whether the data collected by the symptom checker was more supportive of one of the physician’s diagnoses? What if the data included questions relevant to both? What number/percentage of cases had more than one physician diagnosis?

2) Patient were excluded “if there was a very significant mismatch between patient and physician concerns for visit”. What number/percentage of cases were excluded by this criteria?

3) In listing the 3 key criteria for quality of studies the authors may wish to add a 4th, data entered by patients on their own symptoms, which is missing from most existing studies of SCs, that use existing or made up cases.

Response to reviewer #2:

1) We have adjusted the relevant paragraph of the methods for added clarity. The paragraph is copied here for clarity: “Some physicians documented multiple diagnoses or issues for the healthcare visit. In these cases, each diagnosis was separately categorized as an emergency, urgency, routine, or home appropriate. For example, a patient encounter with dual diagnoses of “1) vaginal discharge not yet diagnosed and 2) lower back pain” were categorized as routine and home appropriate, respectively, because investigations were ordered by the physician for the vaginal discharge and home-treatment solutions were recommended for the lower back pain. Symptom checker responses were then assessed to determine the most appropriate physician diagnosis. In this specific encounter, the patient selected on the symptom checker that the concern was not related to pain; thus the diagnosis “vaginal discharge not yet diagnosed” was used for the purposes of assessing symptom checker performance.”

2) The number of excluded patients are presented in Figure 1. Additional information is provided in the supplementary data, where further commentary is made about the discrepancies between the patient’s symptom checker responses and notes written by physicians.

3) This was a useful comment, and a sentence was added to the introduction to reflect this.

-Done

2. Please state in your methods section whether you obtained consent from parents or guardians of the minors included in the study or whether the research ethics committee or IRB approved the lack of parent or guardian consent.

-Done

3. In your Methods section, please provide additional information about the participant recruitment method and the demographic details of your participants. Please ensure you have provided sufficient details to replicate the analyses such as: a) the recruitment date range (month and year), b) a description of any inclusion/exclusion criteria that were applied to participant recruitment, c) a table of relevant demographic details, d) a statement as to whether your sample can be considered representative of a larger population, e) a description of how participants were recruited, and f) descriptions of where participants were recruited and where the research took place.

-All of the information was already there, but we have changed some words for added clarity. 

-Please see our cover letter for further comments.

---

## [Decision Letter · Decision Letter 1]

11 Nov 2021

PONE-D-20-39920R1Performance of a New Symptom Checker in Patient Triage: Canadian Cohort StudyPLOS ONE

Dear Dr. Chan,

Thank you for submitting your manuscript to PLOS ONE. After careful consideration, we feel that it has merit but does not fully meet PLOS ONE’s publication criteria as it currently stands. Therefore, we invite you to submit a revised version of the manuscript that addresses the points raised during the review process.

We have now received one review on your revised submission. While the reviewer is largely happy with the revisions, they also raise one concern about the exclusions that it would be helpful to see addressed. Please submit your revised manuscript by Dec 26 2021 11:59PM. If you will need more time than this to complete your revisions, please reply to this message or contact the journal office at plosone@plos.org. Please include the following items when submitting your revised manuscript:A rebuttal letter that responds to each point raised by the academic editor and reviewer(s). You should upload this letter as a separate file labeled 'Response to Reviewers'.A marked-up copy of your manuscript that highlights changes made to the original version. You should upload this as a separate file labeled 'Revised Manuscript with Track Changes'.An unmarked version of your revised paper without tracked changes. You should upload this as a separate file labeled 'Manuscript'.If applicable, we recommend that you deposit your laboratory protocols in protocols.io to enhance the reproducibility of your results. Protocols.io assigns your protocol its own identifier (DOI) so that it can be cited independently in the future. For instructions see: https://journals.plos.org/plosone/s/submission-guidelines#loc-laboratory-protocols. Additionally, PLOS ONE offers an option for publishing peer-reviewed Lab Protocol articles, which describe protocols hosted on protocols.io. Read more information on sharing protocols at https://plos.org/protocols?utm_medium=editorial-email&utm_source=authorletters&utm_campaign=protocols.

We look forward to receiving your revised manuscript.

Kind regards,

Andrea Gruneir

Academic Editor

PLOS ONE

Journal Requirements:

Reviewers' comments:

Reviewer's Responses to Questions

**Comments to the Author**

1. If the authors have adequately addressed your comments raised in a previous round of review and you feel that this manuscript is now acceptable for publication, you may indicate that here to bypass the “Comments to the Author” section, enter your conflict of interest statement in the “Confidential to Editor” section, and submit your "Accept" recommendation.

Reviewer #2: (No Response)

2. Is the manuscript technically sound, and do the data support the conclusions?

Reviewer #2: Yes

3. Has the statistical analysis been performed appropriately and rigorously? 

Reviewer #2: Yes

4. Have the authors made all data underlying the findings in their manuscript fully available?

Reviewer #2: No

5. Is the manuscript presented in an intelligible fashion and written in standard English?

Reviewer #2: Yes

6. Review Comments to the Author

Reviewer #2: Thank you for your responses to my comments. The figure 1 indicates that about 22% of patients were excluded due to "very significant mismatch between patient and physician concerns for visit" which impacts on the overall usability of the system. It is a lot higher proportion than what we have seen in our own studies. Any indication as to why these patients didn't use the symptom checker for the complaint that brought them to the ED or primary care center?

One other comment relates to the question of the algorithm type (raised by reviewer 1). It is very helpful to reviewers and readers to know broadly what type of algorithm is involved. It presumably was not created by any machine learning technique - useful point to clarify. If it uses a Bayesian network for example (like some other symptom checkers) that would be helpful to know.

7. PLOS authors have the option to publish the peer review history of their article (what does this mean?). If published, this will include your full peer review and any attached files.

Reviewer #2: **Yes: **Hamish Fraser

---

## [Author Response · Author response to Decision Letter 1]

11 Nov 2021

Regarding reviewer #2 and PLOSONE editor comments:

Thank you very much for your thoughtful review of our paper!

Regarding "4. Have the authors made all data underlying the findings in their manuscript fully available?",

-the PLOS one editors specifically did not further comment upon this, and we presume that our detailed response from the prior review cycle was satisfactory.

Regarding "Reviewer #2: Thank you for your responses to my comments. The figure 1 indicates that about 22% of patients were excluded due to "very significant mismatch between patient and physician concerns for visit" which impacts on the overall usability of the system. It is a lot higher proportion than what we have seen in our own studies. Any indication as to why these patients didn't use the symptom checker for the complaint that brought them to the ED or primary care center?"

-Unfortunately, we can only hypothesize about why patients did not respond truthfully. Our foremost hypothesis is that patients wanted to participate and use our symptom checker application out of personal interest, but did not want to reveal their own personal health information. We have now noted this hypothesis in the paper's discussion "The physician visit diagnosis and patient’s symptom checker responses were significantly mismatched for 135 patients and excluded from analysis (S1 Table); this may have been because patients wanted to utilize the symptom checker out of personal interest, but did not wish to reveal details about their personal health information."

-In the Supporting Documents, the management of patient data for every individual patient is provided for transparency. Specifically, it does details why a mismatch was flagged for the patient. For example, for patient in row A17, the patient using the symptom checker responded that they were not experiencing any skin related issues or pain; however, their hospital diagnosis was "cellulitis of the hand" which inherently is a skin related issue that almost always presents with pain.

Regarding "One other comment relates to the question of the algorithm type (raised by reviewer 1). It is very helpful to reviewers and readers to know broadly what type of algorithm is involved. It presumably was not created by any machine learning technique - useful point to clarify. If it uses a Bayesian network for example (like some other symptom checkers) that would be helpful to know.":

-A phrase has now been added to clarify that the algorithm was "manually compiled into a decision tree with recursive elements".

---

## [Editor Report · Decision Letter 2]

16 Nov 2021

Performance of a New Symptom Checker in Patient Triage: Canadian Cohort Study

PONE-D-20-39920R2

Dear Dr. Chan,

We’re pleased to inform you that your manuscript has been judged scientifically suitable for publication and will be formally accepted for publication once it meets all outstanding technical requirements.

Kind regards,

Andrea Gruneir

Academic Editor

PLOS ONE
---

## [Editor Report · Acceptance letter]

19 Nov 2021

PONE-D-20-39920R2 

Performance of a New Symptom Checker in Patient Triage: Canadian cohort study 

Dear Dr. Chan:

I'm pleased to inform you that your manuscript has been deemed suitable for publication in PLOS ONE. Congratulations! Your manuscript is now with our production department. 

Kind regards, 

on behalf of

Dr. Andrea Gruneir 

Academic Editor

PLOS ONE